# Mutation bias alters the distribution of fitness effects of mutations

**Mrudula Sane¤, Shazia Parveen, Deepa Agashe** *

National Centre for Biological Sciences, GKVK Campus, Bengaluru, Karnataka, India

¤ Current address: Department of Ecology, Behavior and Evolution, University of California San Diego, San Diego, California, United States of America

* dagashe@ncbs.res.in

## Abstract

Mutation bias is an important factor determining the diversity of genetic variants available for selection. As adaptation proceeds and some beneficial mutations are fixed, new beneficial mutations become rare, limiting further adaptation. The depletion of beneficial mutations is especially stark within the mutational class favored by the existing mutation bias. Recent theoretical work predicts that this problem may be alleviated by a change in the direction of mutation bias (i.e., a bias reversal). If populations sample previously underexplored types of mutations, the distribution of fitness effects (DFE) of mutations should shift towards more beneficial mutations. Here, we test this prediction using *Escherichia coli*, which has a transition mutation bias, with ~54% single-nucleotide mutations being transitions compared to the unbiased expectation of ~33% transitions. We generated mutant strains with a wide range of mutation biases, from 97% transitions to 98% transversions, either reinforcing or reversing the wild-type transition bias. Quantifying DFEs of ~100 single mutations obtained from mutation accumulation experiments for each strain, we find strong support for the theoretical prediction. Strains that oppose the ancestral bias (i.e., with a strong transversion bias) have DFEs with the highest proportion of beneficial mutations, whereas strains that exacerbate the ancestral transition bias have up to 10-fold fewer beneficial mutations. Such dramatic differences in the DFE should drive large variation in the rate and outcome of adaptation, suggesting an important and generalized evolutionary role for mutation bias shifts.

## Introduction

Mutation is the major source of genetic variation, and it is important to quantify the phenotypic and fitness effects of new mutations. A substantial body of work has therefore focused on determining the statistical distribution of mutational effects (the distribution of fitness effects, DFE) and the evolutionary processes that shape the DFE [1–3]. The DFE determines the number and proportion of beneficial mutations, a key

**Data availability statement:** All relevant data are within the paper and its Supporting information files.

**Funding:** This work was funded by the DBT/Wellcome Trust India Alliance (Grant no. IA/S/23/2/506989 to DA), the National Centre for Biological Sciences (NCBS–TIFR) and the Department of Atomic Energy, Government of India (Project Identification No. RTI 4006 to DA), and the University Grants Commission, India (fellowship number 211610044747 to SP). The funders did not play any role in the study design, data collection and analysis, decision to publish, or preparation of the manuscript.

**Competing interests:** I have read the journal's policy and the authors of this manuscript have the following competing interests: DA is a member of PLOS Biology's Editorial Board. The other authors declare that no competing interests exist.

**Abbreviations:** BPS, base-pair substitutions; DFE, distribution of fitness effects; LB, lysogeny broth (growth medium); MA, mutation accumulation; MMR, mismatch repair; Ts, transition mutations; Tv, transversion mutations; WGS, whole-genome sequencing; WT, wild-type.

parameter in population genetic models of evolutionary change. A broad and general understanding of the DFE and factors that influence it is thus crucial to predict adaptation rates, trajectories, and fates of evolving populations. Ultimately, such predictions and their tests are key to tackling problems such as the emergence of antimicrobial resistance and rapid environmental change that threatens populations [4]. From numerous studies using different approaches to estimate or quantify the DFE [5], we know that it is influenced by several factors such as the genetic background, the environment, the effective population size, and prior history of adaptation [1–3,6,7].

New work in the past few years has suggested that the DFE may also vary with the nature of the underlying mutations. The mutation spectrum—describing relative frequencies of different types of mutations—is typically biased towards specific classes of mutations. For instance, most organisms show a bias towards more transition mutations [8]. If different mutational classes have distinct fitness consequences, such pervasive mutation biases may affect the DFE. For example, an *Escherichia coli* strain that samples a higher proportion of AT→CG transversion mutations had a distinct DFE for antibiotic resistance, compared to a strain that samples more GC→TA transversions [9]. During laboratory evolution under increasing antibiotic stress, strains with different mutational bias developed resistance using distinct mutational paths [10]. Expanding this idea to genome-wide mutational biases, we recently showed that changing the mutation bias can alter the *E. coli* DFE across several environments with limiting carbon sources [11]. Specifically, on deleting a DNA repair gene (*mutY*)—which increases the incidence of transversion mutations compared to the wild type (WT)—we observed an ~6% increase in the fraction of beneficial mutations on average across environments. The form of the global DFE may thus change depending on the mutation spectrum of an organism and the fitness effects of different types of mutations.

Simulations of adaptive walks as well as a mathematical model uncovered general conditions under which mutation bias shifts should change the DFE, and the evolutionary impacts of the expected DFE changes [11,12]. This led to a key prediction: opposing (i.e., reversing or reducing) the direction of the ancestral mutation bias should increase the fraction of beneficial mutations (Fig 1). Opposing an existing mutation bias is predicted to be generally beneficial because it allows populations to explore previously under-sampled mutational space, including beneficial mutations that were not fixed (and were therefore available). For instance, consider a population that has evolved in a constant environment with a transition mutation bias (e.g., WT *E. coli*) for some time. It will have gradually sampled, and fixed, many of the possible beneficial transition mutations; but it would have sampled only a small fraction of available beneficial transversions. Thus, adaptation results in a depletion of the beneficial part of the DFE (the DBFE), especially the well-sampled mutational classes [12–14]. On introducing a transversion bias (reversing the existing bias), such a population is more likely to sample beneficial transversions, leading to a right-shifted DFE (Fig 1). In contrast, if the ancestral bias is reinforced (i.e., with a stronger transition bias in *E. coli*), we predict that the DFE should shift left compared to the WT, with a larger fraction of deleterious mutations. Thus, mutation biases may play important roles in shaping DFEs [11,12].

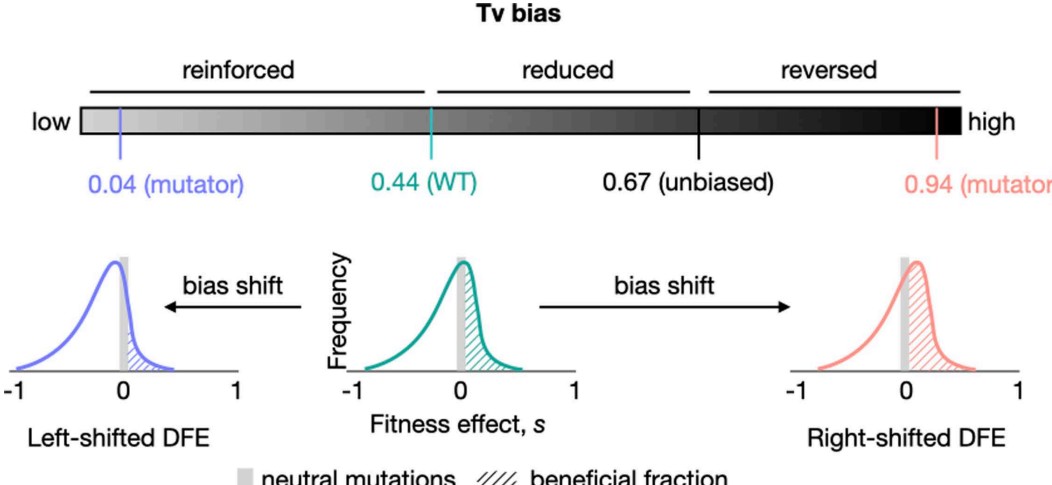

**Fig 1. Schematic of the key prediction tested in this work.** As the transversion (Tv) mutation bias of WT *E. coli* is shifted away from the ancestral bias, the resulting distribution of fitness effects (DFE) is predicted to change. Specifically, it should shift left with a bias reinforcement, with a lower fraction of beneficial mutations. In contrast, reversing the ancestral bias is predicted to cause right-shifted DFEs with higher proportions of beneficial mutations.

Here, we experimentally test these predictions by generating empirical DFEs of new mutations in six *E. coli* strains carrying deletions of various DNA repair genes involved in the mismatch repair (MMR) pathway, the 8-oxo-dGTP repair pathway, or the repair of damaged pyrimidines (Table 1). WT *E. coli* has a significant transition (Ts) bias, whereby only ~46% of single-nucleotide mutations are transversions (Tv) (compared to the null expectation of ~67% Tv; Fig 1). Deleting DNA repair genes leads to a wide range of mutation biases, with transversion biases of 0.03 (i.e., 97% Ts) to 0.98 (i.e., 98% Tv) [8,15]. To obtain single-step mutations for constructing DFEs, we allowed several lineages of each strain to evolve independently in a mutation accumulation (MA) regime (some results from MA experiments for WT and Δ*mutY* were described in [11,16]). Evolution under MA allows nearly all mutations to be sampled (but see [17,18]), allowing us to obtain a random representative sample of mutations available to each strain. We used whole-genome sequencing to identify lineages with a single mutation compared to the respective ancestor, measured the fitness effect of each mutation in two environments (rich Lysogeny Broth (LB) and M9 minimal medium with Glucose), and used these data to construct DFEs. Our results provide strong support for our predictions, demonstrating that mutation spectrum shifts can determine the amount of adaptive genetic variation available to populations.

## Methods

### Bacterial strains

We obtained the WT strain of *E. coli* K-12 MG1655 from the Coli Genetic Stock Centre (CGSC, Yale University), streaked it on Luria Bertani (LB) agar (Miller), and chose one colony at random as the WT ancestor for subsequent experiments. We similarly obtained the mutator strains of *E. coli* (Δ*mutT*, Δ*mutH*, Δ*mutL*, Δ*mutS*, Δ*nth*, Δ*nei*, Δ*mutY*) from the Keio collection of gene knockouts from CGSC (BW25113 strain background [19]). These gene knockouts were made by replacing open reading frames with a Kanamycin resistance cassette, such that removing the cassette generates an in-frame deletion of the gene. The design of gene deletion primers ensured that downstream genes were not disrupted due to polar effects. For each mutator strain, we moved the knockout locus from the BW25113 background into the MG1655 (our WT) background using P1 phage transduction [20]. We then removed the kanamycin resistance marker

**Table 1. Strains used in this study, and their mutational characteristics. For each class of mutations, bias is calculated as (no. of mutations of type A)/(no. of mutations of type A + no. of mutations of type B). Brief information about the function of deleted genes is provided in the footnotes [25].**

| Property | ΔmutS† | ΔmutL† | ΔmutH† | Δnth-neiΨ | WT | ΔmutYɸ | ΔmutTɸ |
|---|---|---|---|---|---|---|---|
| MA lines evolved | 350 | 350 | 350 | 300 | 98 | 430 | 300 |
| MA lines successfully sequenced | 345 | 344 | 346 | 285 | 97 | 424 | 271 |
| Total mutations in sequenced lines[a] | 616 | 603 | 908 | 502 | 426 | 798 | 789 |
| MA lines with single mutations[b] | 91 | 97 | 100 | 102 | 94[c] | 113 | 97 |
| **Mutation rate** (×10⁻⁸) | 1.4 | 1.4 | 2.1 | 0.17 | 0.0091 | 0.083 | 2.3 |
| ±95% CI | ±0.14 | ±0.13 | ±0.14 | ±0.015 | ±0.0028 | ±0.016 | ±0.19 |
| BPS | 508 | 509 | 809 | 482 | 387 | 787 | 783 |
| Indel | 108 | 94 | 99 | 20 | 39 | 11 | 6 |
| **Indel bias** | 0.18 | 0.16 | 0.11 | 0.04 | 0.09 | 0.01 | 0.01 |
| ±95% CI | ±0.040 | ±0.038 | ±0.033 | ±0.023 | ±0.057 | ±0.011 | ±0.010 |
| Coding | 440 | 437 | 710 | 400 | 282 | 684 | 657 |
| Noncoding | 68 | 72 | 99 | 82 | 105 | 103 | 126 |
| **Noncoding bias** | 0.13 | 0.14 | 0.12 | 0.17 | 0.27 | 0.13 | 0.16 |
| ±95% CI | ±0.036 | ±0.037 | ±0.035 | ±0.044 | ±0.088 | ±0.032 | ±0.044 |
| Non-synonymous | 290 | 274 | 433 | 273 | 183 | 489 | 462 |
| Synonymous | 150 | 163 | 277 | 127 | 99 | 195 | 195 |
| **Synonymous bias** | 0.34 | 0.37 | 0.39 | 0.32 | 0.35 | 0.29 | 0.30 |
| ±95% CI | ±0.05 | ±0.051 | ±0.051 | ±0.054 | ±0.095 | ±0.043 | ±0.054 |
| Transitions | 491 | 491 | 774 | 430 | 206 | 69 | 12 |
| Transversions | 17 | 18 | 35 | 52 | 178 | 644 | 771 |
| **Transversion bias** | 0.03 | 0.04 | 0.04 | 0.11 | 0.46 | 0.91 | 0.98 |
| ±95% CI | ±0.019 | ±0.020 | ±0.021 | ±0.036 | ±0.099 | ±0.028 | ±0.015 |
| AT→GC | 339 | 361 | 568 | 22 | 118 | 24 | 775 |
| GC→AT | 162 | 143 | 229 | 438 | 206 | 746 | 5 |
| No change | 7 | 5 | 12 | 22 | 60 | 17 | 3 |
| **GC→AT bias** | 0.32 | 0.28 | 0.29 | 0.95 | 0.64 | 0.97 | 0.01 |
| ±95% CI | ±0.049 | ±0.048 | ±0.048 | ±0.025 | ±0.096 | ±0.017 | ±0.01 |

†Genes involved in the methyl-directed mismatch repair (MMR) pathway. MutS recognizes mismatched base-pairs in double-stranded DNA. MutL binds to DNA-bound MutS, and facilitates interaction with MutH. MutH performs daughter-strand recognition and creates single-strand DNA breaks in the daughter strand at the nearest hemi-methylated site. The breaks are then repaired by exonuclease activity extending up to the mismatch, followed by resynthesis by DNA polymerase III.

ΨGenes involved in the repair of damaged pyrimidines in double-stranded DNA. *nth* encodes Endonuclease III, which recognizes and cleaves N-glycosidic bonds of damaged pyrimidines, leaving abasic sites, and also cleaves the phosphodiester backbone 3′ to the site to create single-strand DNA breaks. *nei* encodes Endonuclease XIII, and shares specificity with Endonuclease III.

ɸGenes involved in the 8-oxo-dGTP repair pathway. MutT hydrolyzes cytosolic 8-oxo-dGTP to 8-oxo-dGMP, preventing its incorporation into DNA during replication. MutY recognizes 8-oxo-G:A mis-pairs in double-stranded DNA and excises the adenine, leaving an abasic site.

[a]Mutations used to calculate mutation rates and spectra.

[b]Lines used for fitness measurements and inference of DFEs.

[c]All of these single-step mutations were not derived from different MA lines (i.e., 80 of the 94 mutations were obtained from WT block 1 that contained only 38 lines, such that some lines contributed multiple single mutants). This is because 46 of them represent second-, third-, or fourth-step mutations occurring in a given lineage, whose fitness effect was estimated with respect to the immediate ancestor (i.e., carrying first-, second-, or third-step mutations, respectively). Details are described in [16], and in S1 Table.

by transforming kanamycin-resistant transductants with pCP20, a plasmid carrying the flippase recombination gene and ampicillin resistance marker. We grew ampicillin-resistant transformants at 42°C in LB broth overnight to cure pCP20 and streaked out 10 μL of these cultures on LB plates. After 24 hours, we replica-plated several colonies on both LB + kanamycin agar plates and LB + ampicillin agar plates, to screen for the loss of both kanamycin and ampicillin resistance. We PCR-sequenced the knockout locus to confirm removal of the kanamycin cassette. For generating the Δnth-nei double knockout, we first created a Δnth strain, and then moved the Δnei locus into this background using P1-phage transduction as described above. In the process of making gene knockouts, all mutator strains (except Δnth-nei) acquired background mutations (S1 Data) due to multiple generations of growth that occurred during the screening process.

## Mutation accumulation (MA) experiments

We used MA experiments of varying length to obtain mutator strains carrying single mutations that would reflect the mutation spectrum of each mutator (~100 per strain). We isolated a single colony of each ancestor, suspended it in LB broth, and plated it to obtain as many colonies as were needed for each independent MA line. We used this same broth culture within 3–4 h of growth to extract DNA for whole-genome sequencing (WGS) (see next section). For mutators with very high mutation rates (ΔmutH, ΔmutT, ΔmutL, and ΔmutS), each MA block had its own ancestor (since each time cells are grown up from freezer stocks, there is a very high probability that new mutations will arise) whose sequence was used to subtract the ancestral mutations from offspring lines (see below). For mutators with intermediate mutation rates and for WT, different MA blocks of a strain were started with the same ancestor.

The MA protocol minimizes the effect of selection, allowing sampling of a wide range of mutational effects, largely independent of their fitness consequences. For each MA line, every 24 h we streaked out a random colony (closest to a pre-marked spot) on a fresh LB agar plate. For MA experiments lasting more than a day, every 4–5 days we inoculated a part of the transferred colony into LB broth at 37°C for 2–3 h and froze 1mL of the growing culture with an equal amount of 60% glycerol at –80°C. For 1-day MA experiments, we similarly cultured and froze the final chosen colony. We used these freezer stocks of the MA lines for sequencing. We chose the length and number of replicate lines of the MA experiments for each strain depending on mutation rate and logistical feasibility. Since the WT had a relatively low mutation rate, we founded a small number of MA lines to make daily transfers feasible, but evolved them for many generations. For the mutators, we founded larger numbers of MA lines but evolved them for just a few generations. We also split the large number of MA lines into blocks (except ΔmutT; S1 Table) to make transfers and handling easier. Thus, most MA experiments were performed across at least two experimental blocks (S1 Table).

We founded multiple MA lines of each strain from single colonies: WT (98 lines), ΔmutT (300 lines), ΔmutH (350 lines), ΔmutL (350 lines), ΔmutS (350 lines), ΔmutY (430 lines), and Δnth-nei (300 lines) and propagated them through daily single-colony bottlenecks on LB agar plates (Table 1). We previously showed that our WT strain goes through ~27 generations in 24 h of growth on LB agar [16]. We used this estimate to calculate the number of generations elapsed in our MA experiments. We evolved WT lines in two experimental blocks: one block of 38 lines evolved for 300 days (8,250 generations, described previously in [11,16]), and a second block of 60 lines evolved for 85 days (2,295 generations) (S1 Table). We evolved mutators with intermediate mutation rates in short MA experiments: Δnth-nei in two blocks (block 1: 80 lines, 8 days, ~216 generations; block 2: 220 lines, 8 days, ~216 generations), and ΔmutY in three blocks (block 1: 300 lines, 12 days, ~314 generations, described previously in [11]; block 2: 80 lines, 5 days, ~135 generations; block 3: 50 lines, 1 day, ~27 generations; S1 Table). Finally, we evolved mutators with very high mutation rates in very short MA experiments: ΔmutS (block 1: 300 lines, 1 day, ~27 generations; block 2: 50 lines, 1 day, ~27 generations), ΔmutL (block 1: 300 lines, 1 day, ~27 generations; block 2: 50 lines, 1 day, ~27 generations), ΔmutH (block 1: 300 lines, 1 day, ~27 generations; block 2: 50 lines, 1 day, ~27 generations) and ΔmutT (block 1: 300 lines, 1 day, ~27 generations; S1 Table).

## Whole-genome sequencing to identify clones with single mutations

We sequenced individual colonies from the MA experiments to identify all clones carrying a single mutation relative to their ancestor. For WT, Δ*nth-nei* and Δ*mutY*, we inoculated 2 μL of the frozen stock of each evolved MA line into 2 mL LB broth, and allowed the cells to grow overnight at 37°C with shaking at 200 rpm. For Δ*mutT*, Δ*mutH*, Δ*mutL*, and Δ*mutS*, we allowed cells from frozen stocks to only grow for 3–4 h, to minimize the accumulation of additional mutations. Next, we extracted genomic DNA (GenElute Bacterial Genomic DNA kit, Sigma-Aldrich) and quantified it using the Qubit HS dsDNA assay kit (Invitrogen). We prepared paired-end libraries from each line and the respective MA ancestors, and sequenced them on an Illumina platform (either 2 × 100 bp, or 2 × 125 bp, or 2 × 250 bp). S1 Table provides details of the library preparation and sequencing methods used, as well as the sequencing depth achieved for each strain. WGS for some MA lines was unsuccessful, i.e., we obtained a very small number of reads or no reads at all; these lines were excluded from further analyses (Tables 1 and S1).

For each sample where WGS was successful, details of the number of mutations called are given in S2 Data. We aligned reads with average quality score > Q30 to the NCBI reference *E. coli* K-12 MG1655 genome (RefSeq accession NC_000913.2) using the Burrows-Wheeler short-read alignment tool BWA [21], and used SAMtools to further process the BWA outputs and generate pileup files [21]. Next, we used the default parameters in the VarScan package [22] to extract lists of base-pair substitutions and short indels (<10-bp length). We used Breseq with default parameters [23] to identify long indels and duplications. From these mutation lists, we only retained mutations that satisfied the following three criteria: (i) mutations represented on both the plus and the minus strand, (ii) mutations supported by at least 4 reads per strand, and (iii) mutations with frequency > 80%. The first two filters would remove mutations with weak support, and the last filter would remove mutations that may have arisen either during the late stages of colony growth in MA experiments, or during the brief period of growth for stock preparation or DNA extraction. We performed this filtering using custom scripts written in R and Python. Finally, we used a custom R script to remove mutations present in the corresponding ancestor from the mutation list of each evolved line, and generated the final mutation list for each lineage (S3 Data). All scripts used for these analyses are available as Supporting information (S1–S7 Scripts).

We used several measures to maximize and estimate the accuracy and reliability of mutation-calling. To identify ancestral mutations, we sequenced ancestral MA clones at higher depth (S1 Table), and relaxed the 80% frequency filter such that we captured ancestral mutations segregating at lower frequency. This was especially important for strains with very high mutation rate. We confirmed that all offspring MA lines seeded by a given ancestral clone showed the expected set of ancestral mutations at very high frequency. To quantify the potential effects of secondary low-frequency mutations on our fitness measurements (and therefore the DFE), for each evolved MA line, we generated a separate list of mutations after relaxing the 80% frequency filter. We used the number and frequency of these secondary mutations to estimate the robustness of our results (described in the Results section). To determine the false negative rate of our WGS pipeline, we measured the recall rate of two known mutations in our WT ancestor (the progenitor of all our mutators) relative to the NCBI reference genome (RefSeq accession NC_000913.2) [16]: a G → A SNP at position 2845011 and a 2-bp CG insertion at position 4296380, expecting these mutations to be called at 100% frequency in all evolved MA lines. Finally, to determine the false positive rate, we measured the fitness of a subset of single-mutation clones of WT across two growth cycles, expecting a strong positive correlation if the identified mutations were real.

## Estimating mutation rate, spectra, beneficial supply, and deleterious load

We estimated the mutation rate ($\mu$) and mutation bias for each mutator and WT using mutations called from all sequenced isolates (Table 1, row "MA lines successfully sequenced"). We calculated $\mu$ (per bp per generation) as:

$$\frac{Total\ number\ of\ mutations\ in\ the\ strain}{(Genome\ size)(Total\ number\ of\ generations)}$$

The number of mutations are given in Table 1, and the genome size is 4.6 × 10⁻⁶. The number of generations was calculated as:

$$(Number\ of\ days\ of\ evolution)(generations\ per\ day)(number\ of\ lineages)$$

We tested the observed frequency distribution of the number of mutations called in each lineage, against the expected Poisson distribution for random mutations. We calculated mutational biases from the different types of mutations observed in the MA-evolved lines (Table 1) as described in [11]. For instance, we calculated the Tv bias as:

$$\frac{Number\ of\ transverions}{Number\ of\ transitions + Number\ of\ transversions}$$

We estimated 95% confidence intervals as 1.96 times the standard deviation of calculated bias. We estimated the confidence intervals for mutation rate as the known mean mutation rate ± the margin of error for a t-distribution with known mean and unknown standard deviation.

For each strain, we used the estimated mutation rates and the fractions of beneficial and deleterious mutations ($f_b$ and $f_d$, see below) to calculate the predicted total supply of beneficial mutations as:

$$S_b = f_b.\mu.genome\ size$$

and the total genetic load due to deleterious mutations as

$$L_d = f_d.\mu.genome\ size$$

We estimated $S_b$ and $L_d$ using either the WT DFE (i.e., assuming that DFEs were invariant across strains), or using strain-specific DFEs measured in each environment. In each case, we also estimated confidence intervals as 1.96 times the standard deviation of $S_b$ or $L_d$. We then compared these values for each strain, to quantify the impact of mutation bias on the supply of beneficial mutations and the deleterious load.

**Growth rate measurements to construct single-mutation DFEs**

From the complete set of all MA-evolved lines with WGS, we focused on those that had a single new mutation compared to the respective ancestor. This included 91 clones of ΔmutS, 97 ΔmutL, 100 ΔmutH, 102 Δnth-nei, 94 WT (80 from block 1 [16] and 14 from block 2), 113 ΔmutY (79 from block 1 [11], 26 from block 2, and 8 from block 3), and 97 ΔmutT MA lines (Table 1, row "MA lines with single mutations"). We performed all fitness assays and subsequent analyses (described below) with these 694 isolates. We measured growth rates of each evolved MA line with a single mutation and its respective ancestor in two liquid culture media: LB broth (Miller, Difco) or M9 minimal salts (Difco) + 5 mM glucose. We inoculated each isolate from its freezer stock into either LB broth or M9 minimal salts medium with 5 mM glucose, and allowed it to grow at 37°C with shaking at 200 rpm for 14–16 h. We inoculated 2 μL of this culture into 200 μL growth media in 96-well plates (Costar) and incubated the well plate in a Tecan F200 Multimode plate reader at 37°C with orbital shaking at 185 rpm for 16–18 h. Every 15 min, the plate reader measured the optical density (OD600) for all wells. In each plate, we included the MA ancestors relevant for the evolved MA isolates to enable fitness calculations, a reference strain (the parent WT strain) to enable checks for consistency across plate reader runs, and blank control wells to check for media sterility. For each evolved isolate, we used the average growth rate of three technical replicates to calculate the relative growth rate as:

$$\frac{Growth\ rate\ of\ evolved\ strain}{Growth\ rate\ of\ ancestral\ strain}$$

For WT MA lines, we used the WT ancestor; and for mutator MA lines, we used the mutator ancestor. We estimated maximum growth rate, obtained from a linear fit to log OD600 versus time curves, using the Curve Fitter software [24]. The fitness effect of each mutation (*s*) was then calculated as:

$$Relative\ growth\ rate - 1 = \frac{Growth\ rate_{evolved} -\ Growth\ rate_{ancestor}}{Growth\ rate_{ancestor}}$$

We used *s* values of mutations to construct strain- and environment-specific distributions of fitness effects (DFE). Importantly, we corrected our DFEs for the expected selection bias in bacterial colonies in MA experiments as described before [11,17]. Briefly, the correction involves binning the measured selection coefficients into discrete bins, and then down-weighting the beneficial fraction of the DFE and over-weighting the deleterious fraction to account for the slightly higher probability of finding beneficial mutations. The bias-corrected versus uncorrected DFEs are shown in S7 and S8 Figs; note that the bias correction procedure leads to discretized bins in the corrected frequency distributions (DFEs).

## Results

### Accuracy of mutation calling and fitness measurements

As described above, our main goal in this study was to construct and compare single-mutation DFEs across strains with distinct mutation biases. To maximize the accuracy of mutation calling, we used stringent sequencing quality filters (see Methods), e.g., only using lineages with high sequencing depth (mean ~40× across all lines, S1 Table). We also conducted several analyses to test the accuracy and reliability of our WGS pipeline. We first confirmed that two known mutations in our WT ancestor (the parent of all our strains) were called in every single evolved MA line with high frequency (SNP at ~100% allele frequency, indel with >93% read support; S1 Fig), indicating a very low false negative rate. Second, in all strains, the observed frequency distribution of the number of mutations called in each lineage was indistinguishable from the expected Poisson distribution for random mutations (S2 Fig), suggesting that non-random processes or mutation calling protocols did not significantly influence the outcome of our MA experiments. Third, we identified one-mutation clones only when they had a single mutation at >80% frequency; on relaxing this filter we found either no secondary mutations or secondary mutations segregating at low frequencies (S3 Fig). Together, these results suggest high accuracy of mutation-calling in our study.

Next, we tested the repeatability and reliability of our fitness estimates, which entailed measuring exponential growth rates of each of the single-mutation clones. In previous work, we had found high repeatability of growth rates [11,16]. We re-confirmed this for the current study: for a subset of isolates, relative fitness measured by two different experimenters in different years was strongly positively correlated (S4 Fig), as were fitness estimates in 48- versus 96-well plates performed in two different years (S5 Fig). These analyses gave us confidence that our measured fitness values are generally robust. Finally, for a subset of clones of WT, fitness values across two successive growth cycles were also strongly positively correlated (S6 Fig). Together, these results indicate that the single mutations that we called were "real", and that our fitness measurements were reliable.

### Transversion-biased strains have a right-shifted distribution of fitness effects (DFE) with a higher proportion of beneficial mutations and lower deleterious load

To test our hypothesis that transversion-biased *E. coli* strains can access more beneficial mutations, we constructed single-mutation DFEs for strains with different transition/transversion biases (Table 1). Each strain acquired distinct mutations, with no shared mutations (S3 Data). For each strain, we characterized the fitness effects of a total of ~100 single mutations in two growth media: one rich (LB) and one relatively poor (minimal medium with glucose), using exponential growth rate as a measure of fitness. Using these fitness estimates, we first constructed raw DFEs, and then corrected

them to account for selection bias during MA (S7 and S8 Figs). Note that the bias correction does not alter the selection coefficients measured for each mutation, but directly modifies the DFE to down-weight the proportion of beneficial mutations (see Methods). The median selection coefficient of single mutations (estimated from the corrected DFEs) varied from −0.175 to +0.025 (i.e., a 17.5% reduction to 2.5% increase in growth rate) in LB, and from −0.013 to +0.063 (i.e., 1.3% reduction to 6.3% increase in growth rate) in glucose.

As predicted, in both environments, strains that reinforced the WT (ancestral) mutation bias (i.e., were strongly Ts-biased: Δ*mutS*, Δ*mutL*, Δ*mutH*, and Δ*nth- nei*) had left-shifted DFEs relative to WT (Figs 2 and 3). In contrast, the DFEs of strains that opposed the WT mutation bias (i.e., had a strong Tv bias, Δ*mutY* and Δ*mutT*) had relatively right-shifted DFEs. The DFE differences across strains were reflected in the proportion of beneficial mutations ($f_b$), which was significantly higher in Tv-biased versus Ts-biased strains (Fig 4A; S2 and S3 Tables). Concomitantly, the fraction of deleterious mutations ($f_d$) was significantly lower in Tv-biased strains (Fig 4A; S2 and S3 Tables). These patterns did not change when we imposed more stringent conditions for single-mutation calling, e.g., if we constructed DFEs using only clones with no secondary mutations or with low-frequency secondary mutations, or clones with two mutations (S9 Fig). Similarly, reducing sample sizes commensurate with the stringent filtering did not alter the patterns (S10 Fig). Thus, the results are robust, and support our main prediction (Fig 1) that mutation bias shifts that reinforce the Ts bias of WT *E. coli* will have left-shifted DFEs and those that reverse the bias (i.e., are transversion biased) will have right-shifted DFEs (Fig 4A). A related prediction is that the magnitude of the DFE shift is correlated with the magnitude of the bias shift; our results are also consistent with this prediction (S11A and S11B Fig, $f_b$ increases significantly with increasing Tv bias; S11C and S11D Fig, $f_d$ tends to reduce with Tv bias, but not significantly so). However, we caution that given the limited number of bias-shifted strains in our dataset, additional work is required to adequately test this correlation. Despite the overall trends described above, we note some exceptions. In LB, Δ*mutY* had a much higher $f_b$ compared to Δ*mutT*, despite a slightly weaker Tv bias (Fig 4A and 4B; S2 Table). In glucose, Δ*mutL* and Δ*mutH* had higher $f_b$ values despite the same transition bias as Δ*mutS* and a stronger Ts bias than Δ*nth-nei* (S3 Table). We explore these exceptions in more detail in the Discussion section.

An important consequence of increased genomic mutation rates is that mutators enjoy an increased total beneficial mutation supply ($S_b$), but must also contend with higher deleterious genetic load ($L_d$)—both important factors in determining their fates during adaptation [26–28]. We first calculated $S_b$ and $L_d$ for all strains assuming WT $f_b$ and $f_d$ (i.e., assuming similar DFEs across strains) [29]. As expected, these values scale linearly with the genomic mutation rate, with strains with 100× higher mutation rates predicted to have a 10- to 250-fold greater beneficial supply and lower deleterious load compared to WT (S12 Fig, S4 and S5 Tables). However, on accounting for the altered $f_b$ and $f_d$ values of mutator strains due to their DFE shifts, we observed deviations from this relationship. Ts biased strains typically had lower $S_b$ while Tv biased strains had higher $S_b$ than expected (S12A and S12B Fig, compare open circles versus filled circles; S4 Table). In the case of $L_d$, accounting for the observed DFEs caused relatively small changes for Ts biased strains, whereas Tv biased strains had a substantially lower $L_d$ than expected based on mutation rate alone (S12C and S12D Fig, S5 Table). The effect of using the empirically observed DFEs was strongest for Δ*mutT*, where the beneficial supply relative to WT increased from ~250-fold to ~650- and ~400-fold in LB and glucose, respectively (S4 Table), and the deleterious load reduced from ~250-fold to ~53-fold in LB and ~58-fold in glucose (S5 Table). Thus, mutation bias shifts could have very large effects on the evolutionary fate of mutators.

## DFE shifts and fraction of beneficial mutations are strongly associated with reversal of Tv bias, with some unexplained variation

The strains used in our analysis differed in several aspects of their mutation spectra (Table 1), so we examined variation in each aspect of the spectrum in more detail (Fig 4B–4G). The increase in $f_b$ across strains was positively correlated with the magnitude of Tv bias (i.e., stronger bias reversal) (S11A and S11B Fig; also compare Fig 4A and 4B). In contrast, no

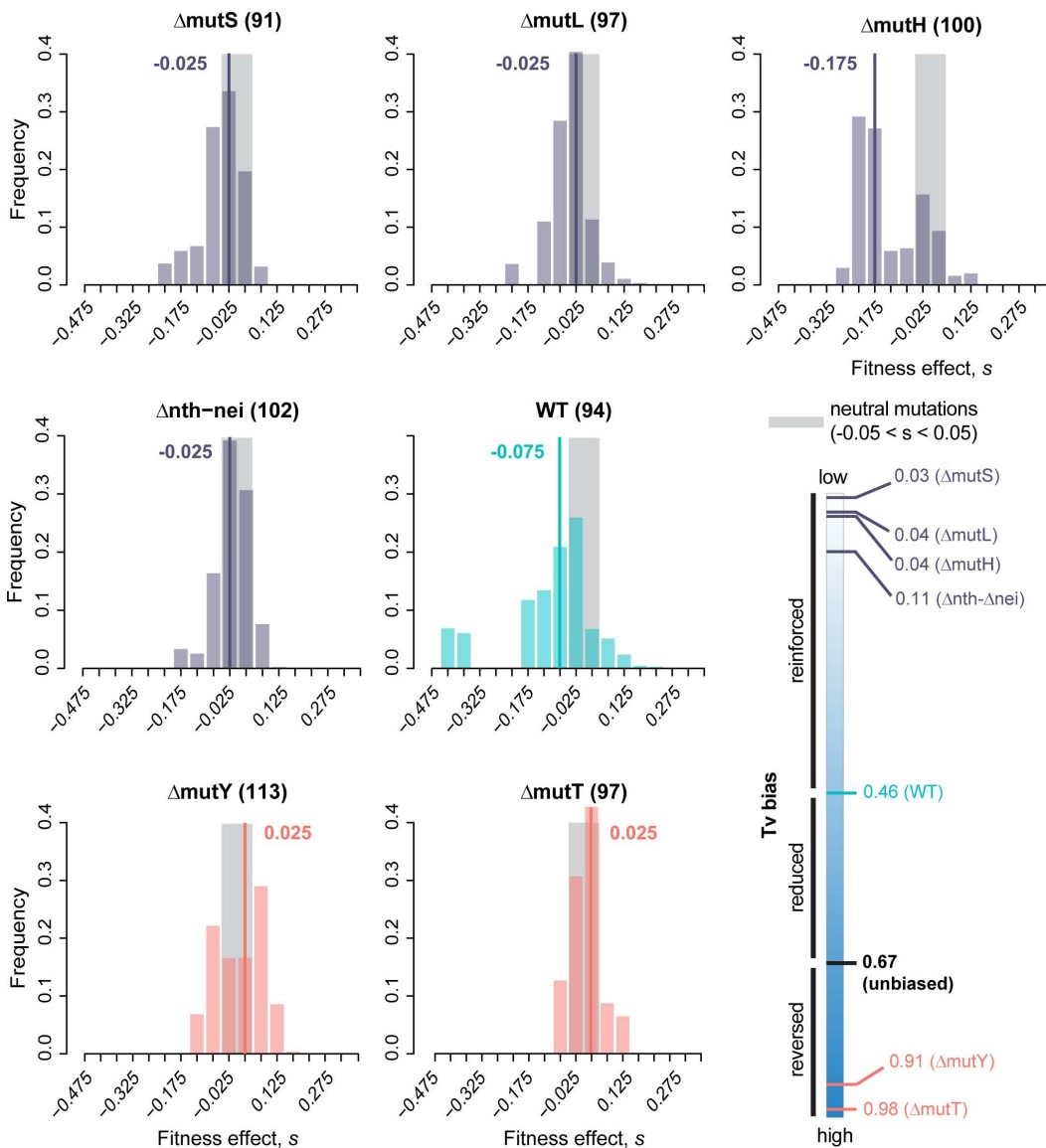

**Fig 2. Distribution of fitness effects of new mutations in a rich medium (Luria Bertani broth, LB) for *Escherichia coli* strains with varying mutation bias.** The distribution of fitness effects of single randomly occurring mutations in each strain (sample size in parentheses), calculated as maximum growth rate relative to the respective ancestor (x-axis). The schematic at bottom right shows the transversion (Tv) bias of each strain. The WT strain is shown in cyan; strains that reinforce the WT transition bias are in purple; strains that reduce or reverse the WT bias (i.e., have a transversion bias) are in pink. Each DFE was corrected for selection bias that could occur during MA; raw (uncorrected) DFEs are shown in S7 Fig; all raw fitness values are provided in S3 Data. Gray areas indicate neutral mutations ($s = 0 \pm 0.05$ to account for experimental measurement error); bold lines and numbers indicate median values of s. Data underlying this figure are given in S4 Data.

other axis of variation in mutation spectrum was correlated with variation in $f_b$ across strains ($p > 0.05$ in each case; compare Fig 4A with Fig 4C–4F). In each of these cases, either the range of variation in mutation bias across strains was very small (indel bias, synonymous bias, non-coding bias; Fig 4C–4E, Table 1) or there was no consistent pattern (noncoding bias and GC→AT bias; Fig 4D and 4F, respectively). Even when we considered each type of base substitution separately, the fitness effects and type of mutation were not associated (Fig 4G). For instance, Δ*mutY* and Δ*mutT* each sample

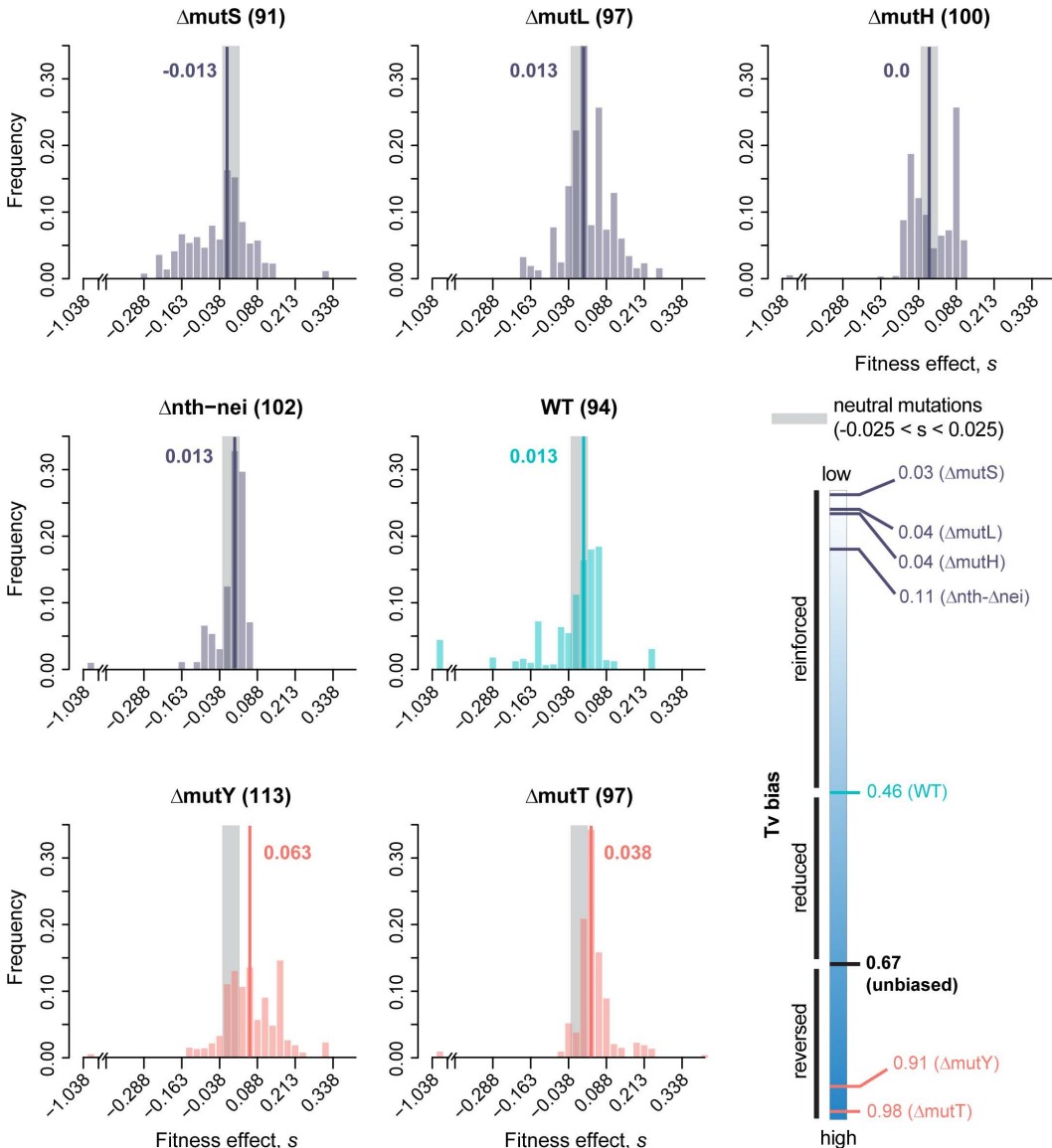

**Fig 3. Distribution of fitness effects of new mutations in a minimal medium (M9 broth with 5 mM glucose) for *Escherichia coli* strains with varying mutation bias.** The distribution of fitness effects of single randomly occurring mutations in each strain (sample size in parentheses), calculated as maximum growth rate relative to the respective ancestor (x-axis). The schematic at bottom right is identical to the schematic in Fig 2, and shows the transversion (Tv) bias of each strain. The WT strain is shown in cyan; strains that reinforce the WT strain's transition bias are in purple; strains that reduce or reverse the WT bias (i.e., have a transversion bias) are in pink. Each DFE was corrected for selection bias that could occur during MA; raw (uncorrected) DFEs are shown in S8 Fig; raw fitness values are provided in S3 Data. Gray areas indicate neutral mutations ($s = 0 \pm 0.025$ to account for experimental measurement error); bold lines and numbers indicate median values of $s$. Data underlying this figure are given in S5 Data.

distinct types of transversion mutations, yet both have the highest $f_b$ values in both environments. Finally, pooling data across all strains in our dataset, transversion mutations were significantly more beneficial than transitions (Fig 5A and 5B), whereas no such fitness difference was observed for any other aspect of the mutation spectrum (except BPS versus indels in glucose; S13 Fig). Together, these results pointed to transversion bias as the dominant cause of DFE differences across strains.

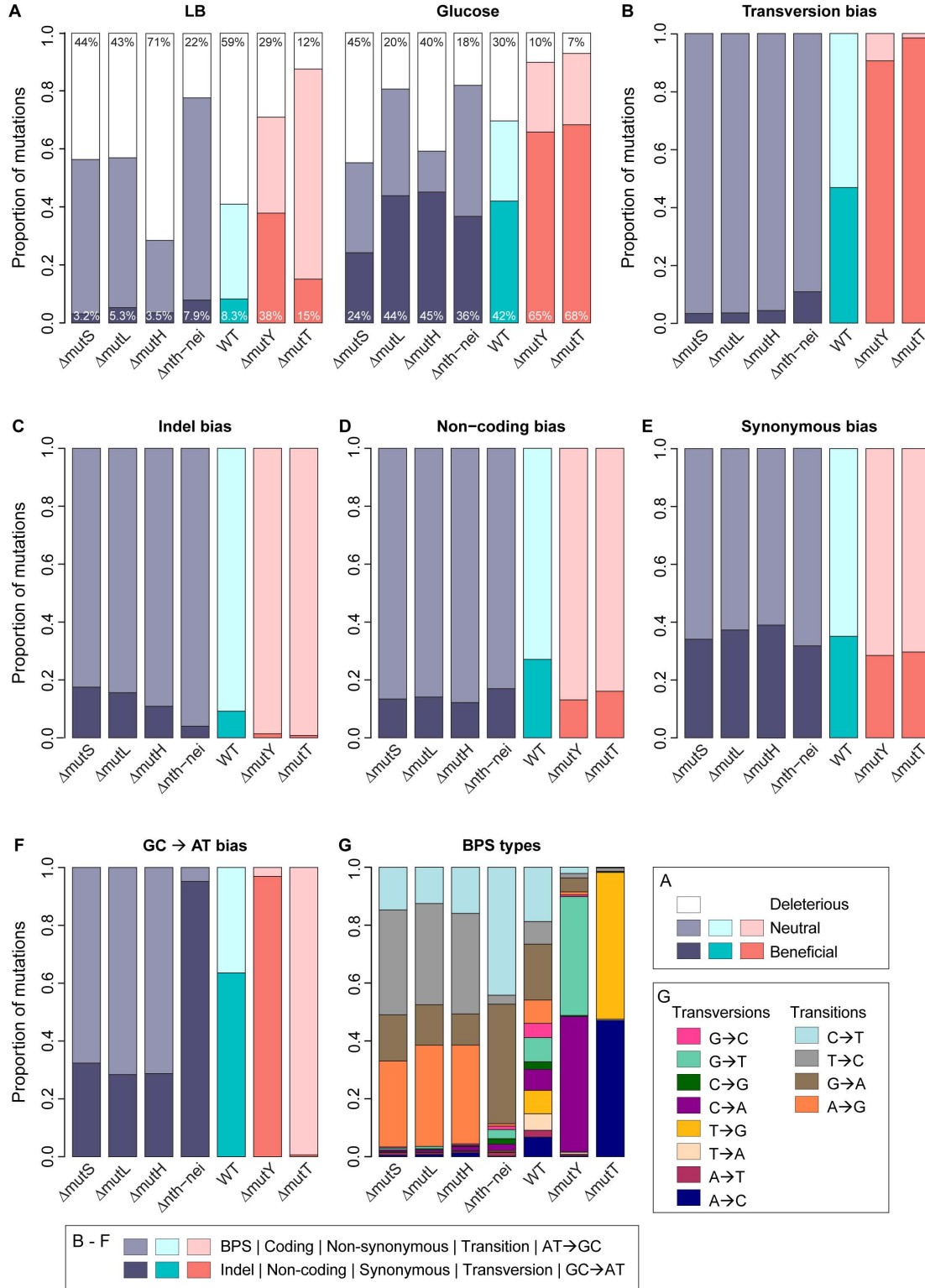

**Fig 4. Differences in DFEs of strains are most strongly associated with transversion bias.** In panels **A—F**, bars are colored by the mutation bias of each strain, as indicated in Figs 2 and 3. **(A)** Stacked bar plots show the total fraction of neutral, deleterious, and beneficial mutations observed in the DFEs of each strain in each growth medium, extracted from the DFEs shown in Figs 2 and 3. Percentage deleterious and beneficial mutations are

indicated at the top and bottom of each bar, respectively. Panels B–G show different aspects of the mutation spectra of strains, with darker vs. lighter shades indicating mutational classes. **(B)** Tv bias **(C)** Indel bias **(D)** Noncoding mutation bias **(E)** Synonymous mutation bias **(F)** GC→AT bias (note that mutations that do not affect GC→AT bias, i.e., AT→TA and GC→CG mutations, are not shown here) and **(G)** Types of base-pair substitutions (BPS). Data underlying this figure are given in S6 Data.

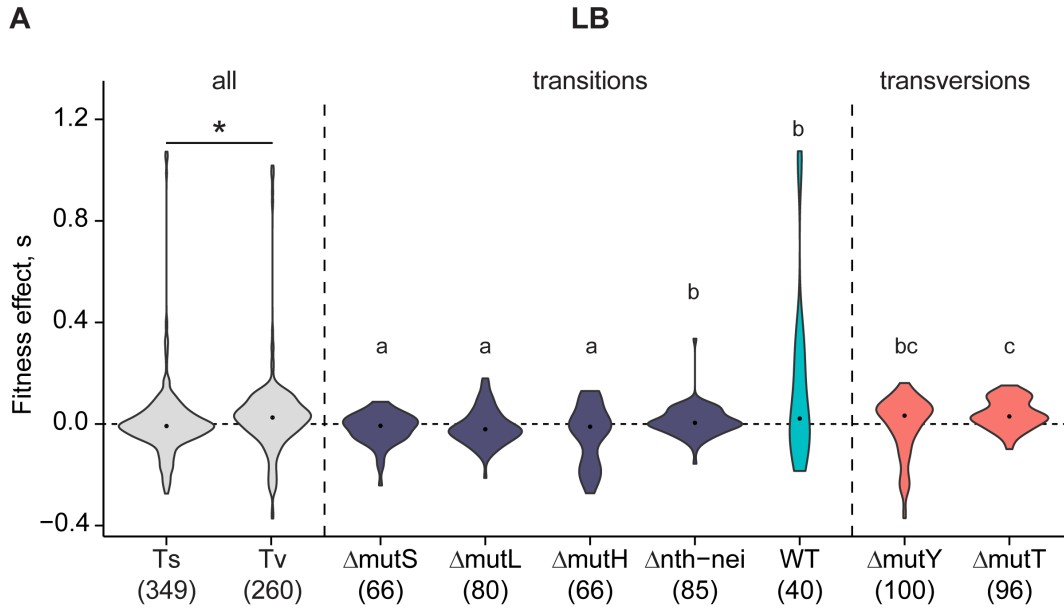

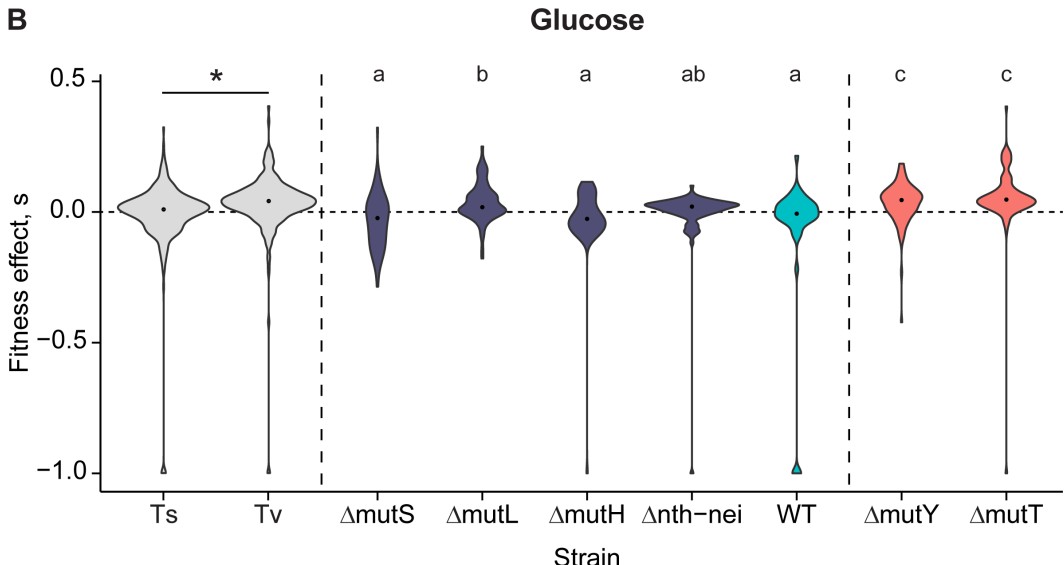

**Fig 5. Transversion mutations are more beneficial than transition mutations.** Fitness effects of mutations in **(A)** LB and **(B)** glucose; sample sizes are shown in x-axis labels in panel A. Note that y-axis ranges differ across panels. In each panel, the first two boxplots compare the effects of all transition vs. transversion mutations, pooled across all strains (asterisks indicate significant differences in Wilcoxon's rank-sum tests; LB: $p = 7.7E-10$; Glucose: $p = 5.1E-10$). Following this, boxplots show the fitness effects of only Ts mutations from transition-biased strains and only Tv mutations from transversion-biased strains. Strains with the same letter have similar fitness effects based on pairwise Wilcoxon's rank sum tests with Benjamini–Hochberg correction (i.e., all strains marked "a" have similar fitness, and all are significantly different from strains marked with "b" or "c". Strains marked "bc" are similar to both "b" and "c"). Comparisons across all other types of mutations are shown in S13 Fig. Data underlying this figure are given in S7 Data.

Next, we tested whether the observed associations between transition/transversion bias and fitness effects of mutations are confounded by strain background (i.e., specifically which DNA repair genes were deleted and what other mutations occurred during the genetic manipulations (S1 Data). We first tested whether higher initial fitness of each strain was associated with lower $f_b$, reflecting the expected pattern of diminishing beneficial mutations with increasing background fitness [30,31]. Although the fitness of the original deletion strains varied significantly in both media, it was not correlated with $f_b$ (S14 Fig). We then conducted pairwise comparisons between strains, considering only Ts and Tv mutations. We expected that Tv mutations should be generally more beneficial than Ts mutations regardless of strain background, and the same type of mutation (Ts or Tv) should have similar fitness effects in all strain backgrounds. These predictions are broadly borne out in both media (Fig 5A and 5B): Tv mutations in Δ*mutY* and Δ*mutT* had similar (and higher) *s* values than Ts mutations, though in LB the fitness effects of Tv in Δ*mutY* were similar to Ts in WT and in Δ*nth-nei*.

Notably, in some cases Ts mutations in different Ts-biased strains had significantly different fitness effects (Fig 5A and 5B). In LB, Ts mutations in WT and Δ*nth-nei* strains were more beneficial than other Ts-biased strains, and in glucose, Ts in Δ*nth-nei* and Δ*mutL* were more beneficial. These exceptions could potentially be explained by aspects of the mutation spectra other than Ts/Tv bias, but these strains do not stand out as exceptional along other axes of the mutation spectrum (Fig 4B–4G, Table 1). Further, comparing Ts-biased strains in each growth medium, we found very few and inconsistent differences in fitness effects along other axes of mutation bias (e.g., coding versus non-coding, synonymous versus non-synonymous; S6 Table), indicating that they cannot explain the differences in fitness effects across Ts-biased strains. To summarize, the broad patterns of DFE variation that we observed are explained by Ts/Tv bias, but there is additional variation among Ts-biased strains that remains unexplained.

### Reversal of the GC/AT bias does not alter DFEs

Prior simulations had predicted that the beneficial effects of sampling unexplored mutational space should extend to any axis of the mutation spectrum, including the GC→AT bias [11]. However, here we did not observe the predicted impacts of reversing the GC→AT bias. The WT has a slight GC→AT bias relative to the unbiased expectation of 0.5, and all three MMR strains as well as Δ*mutT* reverse this bias, whereas Δ*mutY* and Δ*nth-nei* strongly reinforce the bias (Table 1, Fig 4F). However, as discussed above, the DFEs of the strains were not correlated with the magnitude of GC→AT bias, and overall, the fitness effects of GC→AT mutations versus AT→GC mutations were not distinguishable (Fig 6A and 6B). Since the GC→AT bias varies substantially across genes (S15A and S15B Fig), we hypothesized that local rather than global bias reversal may be more relevant. However, mutational effects were not correlated with gene GC content regardless of the direction of mutation bias (Fig 6C and 6D; also see S15C–S15F Fig). Thus, in contrast to the effects of reversing the Tv bias and contrary to our previous simulations, neither local nor global GC bias reversal altered the fitness effects of new mutations.

### Discussion

Our results represent the first systematic experimental analysis of the fitness consequences of varying mutation bias, and support the prediction [11,12] that a reversal of an ancestral mutation bias (here, Ts bias in *E. coli*) can lead to a large increase in beneficial mutations. The difference is stark, with Tv-biased strains showing ~2.5 to 12 times higher $f_b$ relative to Ts-biased strains (Fig 4A). Note that even a small difference in $f_b$ may alter the dynamics of adaptation in large asexual populations, by increasing the overall beneficial mutation supply (e.g., [32]). These results substantially expand upon our prior work, where we observed significantly higher $f_b$ in Δ*mutY* compared to WT in 9 of 16 carbon environments [11]. One unexplained result in the previous study was the lack of significant differences in $f_b$ in five environments, including LB and minimal glucose media. However, in our current analysis with increased sample sizes (80 versus 94 for WT and 79 versus 113 for Δ*mutY*), we observed significantly higher $f_b$ in both LB and glucose, suggesting that the previously observed inconsistency across environments may have resulted from low statistical power. Another difference between the two studies

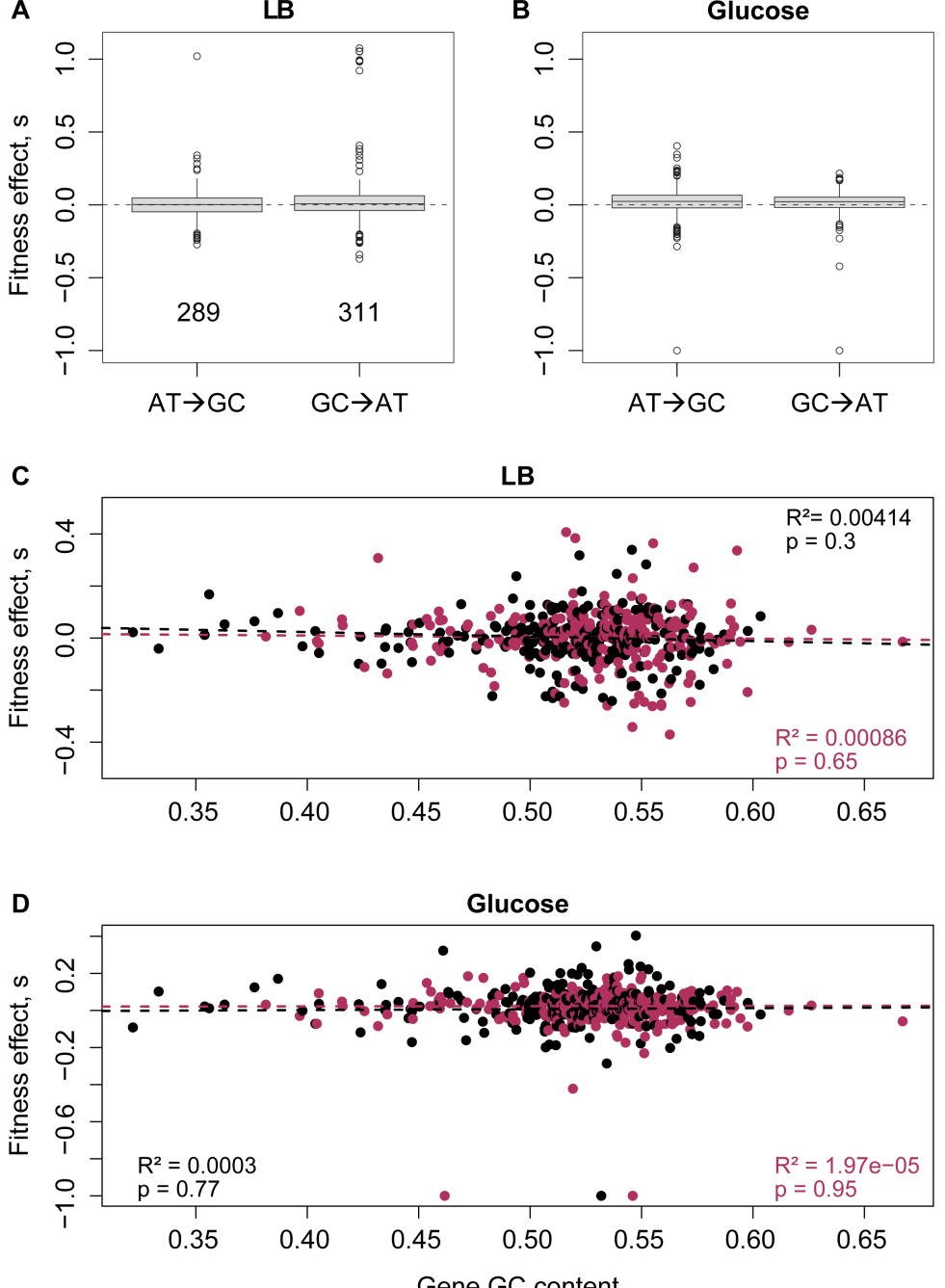

**Fig 6. Reversal of the GC/AT bias does not impact fitness effects of mutations.** Boxplots show fitness effects of AT→GC vs. GC→AT mutations in **(A)** LB and **(B)** Glucose. In each plot, data are pooled across all strains; sample sizes (total number of single mutations tested) are shown in the LB panel. Nine mutations that did not alter GC content (i.e., GC→CG or AT→TA mutations) are not shown here. Mutation type did not change fitness effects in either panel (Wilcoxon's rank-sum test; LB: $p = 0.4$; Glucose: $p = 0.24$). **(C–D)** Scatter plots show the relationship between gene GC content and fitness effects of either AT→GC mutations (maroon points) and GC→AT mutations (black points) in (C) LB and (D) Glucose. Data underlying this figure are given in S8 Data.

is that fitness was measured in slightly different experimental contexts, in 48-well (older study) versus 96-well microplates (current study). However, this cannot explain the different outcomes for WT and Δ*mutY*, because fitness is strongly positively correlated across microplate types (S5 Fig). Thus, we suggest that with sufficiently large sample sizes, a Tv bias should consistently lead to right-shifted DFEs in diverse environments for historically Ts-biased species such as *E. coli*.

More generally, our results add to the growing realization that mutation bias can significantly shape evolutionary dynamics [33–35], by providing direct experimental evidence and suggesting a mechanism through which specific types of bias shifts may alter evolutionary outcomes (i.e., bias reversals lead to right-shifted DFEs with more beneficial mutations). Earlier work had also suggested that bias shifts can dramatically alter evolutionary dynamics, but the underlying mechanism and interpretations were different [9,10]. These studies showed (using some of the same strains used in our work) that distinct mutation spectra endow strains with differential success in sampling specific antibiotic resistance mutations. As a result, some strains are predicted to adapt faster to specific antibiotics. However, this explanation is specific to particular antibiotics and the relevant mutational targets of resistance. Thus, predicting the relative success of different strains in a given antibiotic would require knowledge of resistance mechanisms and whether they are more likely to arise via a specific type of mutation. In contrast, we provide a more general explanation, predicting that the effect of mutation bias depends on the prior evolutionary history of the population, such that a large shift favoring a previously poorly sampled class of mutations should be generally advantageous [11,12]. We hope that future work will test the effect of specific matches between selection and mutation bias versus a broad bias reversal, independent of the source of selection. Such analyses are critical to expand our ability to predict adaptive outcomes and mutation fates under diverse selection pressures.

Although our experiments support our prediction about the impact of mutation bias reversals on the DFE, there are some interesting points of divergence. For instance, in LB, Δ*mutY* has a larger-than-expected $f_b$, and the reason is not yet clear. Most intriguing is the variation in the DFEs of three strains with near-identical Ts bias in glucose (Δ*mutH*, Δ*mutL*, Δ*mutS*; Fig 4A), where the same DNA repair pathway (MMR) is disrupted. Clearly, mutation bias shifts alone cannot explain these differences. One reason could be that some of the MMR genes have other functions apart from DNA repair, such that their deletion directly influences the fitness effects of new mutations. Alternatively, despite attempts to minimize structural disruptions, some repair gene deletions may have caused regulatory changes in downstream genes, leading to strain-specific epistasis with new mutations. Although we did not observe such strain-specific epistasis for a set of 19 mutations placed in both WT and Δ*mutY* backgrounds [11], further experiments are necessary to test this hypothesis for genes in the MMR pathway, whose deletion leads to distinct DFEs. Another interesting exception to the effect of Tv bias reversal is that while the overall pattern of change in $f_b$ and $f_d$ is consistent in both media (LB and glucose), we do see media–specific effects. In previous work with WT and Δ*mutY*, we had also observed significant differences in DFEs across environments with different carbon sources [11]. While variation in DFEs across environments is not surprising, the mechanisms underlying change in the rank order of $f_b$ value across environments remain unclear. Analyzing the causes of the differences in DFEs across strains with similar mutation bias, as well as across environments, is thus a fruitful avenue for further work.

Another unexplained pattern in our current study is that despite strong reversals and reinforcements of the WT GC→AT bias in some strains, GC→AT versus AT→GC mutations had similar fitness effects, and this aspect of the mutation spectrum did not explain variation in the DFE. This is puzzling because simulations had predicted that reversal of *any* aspect of the mutation spectrum should influence the DFE, specifically shown for GC→AT bias [11]. One potential explanation is that the local GC bias rather than genome-wide GC bias is more relevant (e.g., due to the distinct evolutionary history of different genes). A second related hypothesis is that the fitness effects of GC/AT mutations depend strongly on their impact on gene function (e.g., distinct protein-disruptive effects of GC→AT versus AT→GC mutations), and these effects depend on the genome GC content [36]. However, since we do not see any fitness differences between GC→AT or AT→GC mutations globally or locally (in GC-rich or GC-poor genes), neither hypothesis is supported by our data. A

third possibility is that the GC→AT bias varies more frequently and/or dramatically across evolutionary time, reducing the impact of experimentally introduced GC/AT bias reversals relative to Ts/Tv bias reversals. Prior work shows that GC bias is indeed quite dynamic in the bacterial phylogeny (e.g., [37]), but it is impossible to directly test whether GC bias shifts occur more frequently than Tv bias shifts, because the latter do not leave a genomic signature. However, it is possible to approximately estimate the frequency and direction of bias shifts using the gain and loss of DNA repair genes with known effects on Ts/Tv and GC/AT bias. Using such analysis, we had previously observed that reversals of GC/AT bias occurred more frequently than Ts/Tv in the bacterial phylogeny (see Fig 5C in [11]). Thus, we speculate that more frequent changes in the genomic GC content may explain why GC bias reversals did not impact the DFE, and emphasize the need for further theoretical and empirical work on the relative effects of bias shifts along different axes of the mutation spectrum.

A striking result from our study is the high proportion of beneficial mutations observed in all strains in glucose, and in transversion-biased strains in LB (Fig 4A), consistent with our previous analysis of WT and Δ*mutY* strains during growth in several carbon sources [11]. As discussed previously by us and by others, such high $f_b$ values are not as rare as generally believed, especially during MA studies [11,38]. In a recent review, Bao and colleagues suggest that the widely variable outcomes of fitness decline in MA lines (an indicator of the proportion of deleterious versus beneficial mutations) is not easily explained by organism, genetic background, or test environment, but may arise from complex interactions between these and/or other factors [38]. In our current study, we ruled out several mechanisms that could artificially inflate the observed $f_b$ values: selection bias during MA (we corrected our DFEs for such bias), low ancestral fitness leading to high $f_b$ values [30,31,39] (we did not find a correlation between ancestral fitness and $f_b$), and the impact of specific strain backgrounds (we observe a general effect of Ts versus Tv mutations on $f_b$). We hope that further analyses will clarify this issue.

Our results also inform several other open questions in the field, such as the fitness consequences of different types of mutational classes. Perhaps most interesting is the lack of significant differences in the fitness effects of synonymous and nonsynonymous mutations (S13C Fig), adding to the already substantial body of work suggesting that this distinction is not as large or widespread as expected [40,41]. Most prior studies compared the impact of synonymous and non-synonymous changes in specific genes of interest, so our results complement these studies in showing similar fitness effects across hundreds of mutations across the genome. Similarly, we do not find significant differences in coding versus noncoding mutations (S13B Fig), or AT→GC versus GC→AT mutations (Fig 6A and 6B). Together, our results indicate that while the fitness consequences of each of these types of mutations is probably highly context-specific (i.e., within genes, the fitness effect of mutations varies across sites), such patterns are not observed at the genome-wide scale. Future studies with larger sample sizes conducted in different environmental contexts and with distinct organisms will be valuable to test the generality of our results.

The demonstration of large shifts in the distribution of mutational effects as a result of altered mutation bias has important implications for evolutionary dynamics. For instance, the observation that *E. coli* continues to adapt for 50,000 generations of laboratory evolution [42] begs the question of whether the distribution of beneficial mutations is finite. Our results demonstrating an immediate and substantial increase in $f_b$ following a bias reversal suggest that WT *E. coli* does indeed have a finite and depleted set of beneficial mutations. Whether this is broadly true for most natural populations and species remains to be tested. Regardless, our results suggest that changes in mutation spectra across environments or populations—observed in diverse taxa [43–48]—could lead to distinct DFEs in each case, influencing evolutionary dynamics. A second important implication is regarding the evolution of mutators, which have dysfunctional DNA repair and high mutation rates that can facilitate rapid sampling of beneficial mutations [49]. Most mutators also have distinct mutation biases (e.g., except the WT, all strains used in this study are mutators), but the evolutionary implications of such bias shifts in mutators have been only rarely considered [36]. We suggest that the large DFE shifts that we observe—with concomitant changes in the beneficial mutation supply and deleterious load of mutators—can alter the rate and nature of adaptation in new environments. Accounting for these DFE shifts is thus critical to allow more accurate prediction of the fate of mutators with altered mutation biases in populations

under selection. Indeed, a recent analytical and simulation study predicts that right-shifted DFEs driven by mutation bias shifts can significantly enhance the ability of mutator strains to invade non-mutator populations [12]. Thus, some mutators could gain an additional advantage—over and above the effect of high mutation rate—if their mutation bias opposes that of the ancestor. Mutators are observed frequently in natural microbial populations as well as in clinical settings, where they are often associated with drug resistance [50,51], highlighting the need to further understand the effect of mutation bias changes in mutators. More generally, our work highlights several open questions: how often is the distribution of beneficial mutations limited, how often are bias reversals observed in nature, and how often are bias shifts adaptive? Testing the impacts of mutation bias shifts on evolutionary dynamics is thus a promising direction for future research.

## Supporting information

**S1 Fig. Recall rate of known background mutations.** We tested whether mutations in the background of the MG1655 clone used to construct all mutator ancestors were recovered in all evolved clones, as expected if sequencing was perfectly accurate. Violin plots show the frequency of two background mutations in our WT ancestor (compared to the NCBI reference sequence NC_000913.2) in all re-sequenced MA-evolved clones **(A)** A G→A mutation at position 2854011, and **(B)** An insertion of CG at position 4296830, in evolved mutator MA lines. Values under each violin are the median of the distribution. Data underlying this figure are given in S9 Data.
(TIF)

**S2 Fig. The observed number of mutations per MA line is Poisson-distributed.** In each panel, open circles represent the expected number of mutations per MA line, assuming a Poisson distribution with $\lambda =$ mean number of mutations observed per MA line. Filled triangles show the observed number of mutations per MA line. Results of a goodness-of-fit chi-squared test comparing observed versus expected distributions are given in each panel. When different MA blocks differed in the number of generations evolved (in the case of WT and $\Delta mutY$), and therefore had significantly different mean numbers of mutations per MA line across blocks, we analyzed blocks separately. Data underlying this figure are given in S10 Data.
(TIF)

**S3 Fig. Allele frequencies of mutations called in single-mutation MA lines.** Histograms show allele frequencies of mutations in MA lines included in the single mutation DFEs. In each panel, data are pooled for all MA lines included in the DFE measurements for that strain (number of lines is given in parentheses; in these MA lines, we recovered only one mutation of >80% frequency). Gray bars represent mutations segregating in MA lines at lower frequencies (<80%) and colored bars represent mutations at >80% frequency. Data underlying this figure are given in S11 Data.
(TIF)

**S4 Fig. Fitness measurements performed by different experimenters across years are strongly correlated.** The plot shows a fitted linear regression (dashed line) and associated statistics of the relationship between fitness measurements in Glucose conducted in 96-well plates by two different experimenters in two different years, for a set of 12 WT MA clones carrying single mutations. Data underlying this figure are given in S12 Data.
(TIF)

**S5 Fig. Relative growth rates of single-mutation WT clones are consistent across measurements in 96- and 48-well plates.** Each panel shows the fitted linear regression (dashed line) and associated statistics of the relationship between relative growth rates of 80 WT clones carrying single mutations obtained in 48-well plates (data from [1]) and 96-well plates (this study), in LB and Glucose. Data underlying this figure are given in S13 Data.
(TIF)

**S6 Fig. Growth rates of single-mutation MA clones are strongly correlated across growth cycles.** Heritable, "real" mutations identified during resequencing should have consistent effects across successive growth cycles. The plot shows growth rates of different MA clones in glucose (mean ± standard error). We show growth rates in the first 16-h growth after reviving from frozen glycerol stocks (x-axis) vs. a second 16-h growth cycle initiated using cultures from the first growth cycle (y-axis). The dashed line indicates equivalent growth rates in both cycles. Pearson's correlation coefficient and associated *p*-value are shown. Data underlying this figure are given in S14 Data.
(TIF)

**S7 Fig. Raw and selection bias-corrected DFEs of all strains in LB.** Raw (open bars) and corrected DFEs (filled bars) of single mutations in each strain's MA-accumulated mutations tested in LB. Corrected DFEs are colored as in Fig 2. Gray areas indicate neutral mutations ($s = 0 \pm 0.05$ to account for experimental measurement error). Data underlying this figure are given in S15 Data.
(TIF)

**S8 Fig. Raw and selection bias-corrected DFEs of all strains in Glucose.** Raw (open bars) and corrected DFEs (filled bars) of single mutations in each strain's MA-accumulated mutations. Corrected DFEs are colored as in Fig 2. Gray areas indicate neutral mutations ($s = 0 \pm 0.025$ to account for experimental measurement error). Data underlying this figure are given in S16 Data.
(TIF)

**S9 Fig. Effect of stringent filtering for single mutation calling on DFEs.** The fraction of beneficial, neutral, and deleterious mutations for DFEs constructed from MA-evolved clones filtered based on the presence and frequency of secondary mutations. We applied three sets of filters to clones from each strain, comparing each DFE (after correcting for selection bias during MA) with the original ("current") DFE reported in Fig 4 (1): clones with exactly one mutation and no detectable secondary mutation, even at low frequency (2); clones with a secondary mutation at less than 20% allele frequency (3); and clones with exactly two mutations at any frequency (4). The proportion of beneficial and deleterious mutations is given in each bar. In all cases, chi-squared tests comparing each filtered set of clones with the current DFE, with Benjamini–Hochberg correction for multiple comparisons, showed a lack of significant differences in each case ($p > 0.05$). Data underlying this figure are given in S17 Data.
(TIF)

**S10 Fig. Effect of reduced sample size on DFEs.** The stringent filtering described in S9 Fig reduced the number of mutants used to construct each DFE. To test the effect of reduced sample size, we subsampled the current DFE (1) reported in Fig 4 with the respective sample size of each filtered category of clones shown in S9 Fig. Plots show the results from 100 iterations, with 95% confidence intervals indicated for the beneficial fraction. The sample size (number of clones) is indicated in each bar. Chi-squared tests comparing each filtered set of clones with the current DFE, with Benjamini–Hochberg correction for multiple comparisons, showed a lack of significant differences ($p > 0.05$). Data underlying this figure are given in S18 Data.
(TIF)

**S11 Fig. The relationship between Tv bias and the fraction of beneficial ($f_b$) and deleterious mutations ($f_d$) available to strains.** Each panel shows the linear regression fit (dashed line) and associated statistics of the relationship between Tv bias and **(A, B)** $f_b$ or **(C, D)** $f_d$, in LB (left panels) and glucose (right panels). $f_b$ and $f_d$ values are given in Fig 3A and Tv bias values are given in Table 1. Points are colored as indicated in Fig 2. Error bars represent 95% confidence intervals around the mean. Data underlying this figure are given in S19 Data.
(TIF)

**S12 Fig. The DFE alters the beneficial mutation supply and deleterious load across mutators.** Plots show the **(A, B)** beneficial supply and **(C, D)** deleterious load experienced by the different mutators as a function of mutation rate, in LB (left panels) and glucose (right panels). Strains are colored as in Fig 1 (purple: Ts-biased strains, teal: WT, pink: Tv-biased strains). Filled circles represent supply or load calculated using the $f_b$ values obtained from the observed DFE for each mutator (Fig 4A; $S_b$ and $L_d$ values shown in S4 and S5 Tables, respectively); open circles represent supply or load calculated assuming $f_b$ values derived from the WT DFE (i.e., if all strains had the same DFE; $S_{b(WT\ DFE)}$ and $L_{d(WT\ DFE)}$ in S4 and S5 Tables, respectively). For each strain, mutation rates used for the calculations are given in Table 1. Calculations of beneficial supply and deleterious load are shown in S4 and S5 Tables. Dashed lines represent the best fit linear regression for open circles (i.e., supply or load as a function of mutation rate, assuming identical DFEs). Data underlying this figure are given in S20 Data.
(TIF)

**S13 Fig. Fitness effects of aspect of the mutation spectrum other than Ts/Tv.** Fitness effects of **(A)** BPS vs. Indel mutations, **(B)** Coding vs. non-coding mutations, and **(C)** Synonymous vs. non-synonymous mutations. In each plot, data are pooled across all strains; sample sizes (total number of single mutations tested) are shown in the LB (left) panels. When differences are significant (Wilcoxon's rank-sum tests), $P$-values are given in the appropriate panel. Data underlying this figure are given in S21 Data.
(TIF)

**S14 Fig. The fraction of new beneficial mutations ($f_b$) available to strains does not vary with ancestral fitness.** Plots show the relationship between $f_b$ and mean ancestral growth rates in LB (left) and glucose (right). Horizontal error bars represent variation in ancestral growth rates (mean ± SE) and vertical error bars represent uncertainty in $f_b$ estimates ($f_b$ ± 95% CI). The $R^2$ and $p$-values from a linear regression of $f_b$ ~ mean ancestral growth rate are shown in each panel. Data underlying this figure are given in S22 Data.
(TIF)

**S15 Fig. Mutational effects are not associated with gene GC content.** Histograms show the distribution of gene GC content in **(A)** *E. coli* K-12 MG1655 genome and **(B)** genes with mutations in our dataset. Vertical black lines show medians. Boxplots show fitness effects of AT→GC vs. GC→AT mutations in **(C, E)** low GC content genes (i.e., GC content less than the genome-wide median GC) and **(D, F)** high GC content genes (i.e., GC content greater than the genome-wide median GC) in (C–D) LB and (E–F) Glucose. Mutational effects were not significantly different across any of the categories shown in these plots (Wilcoxon's rank-sum tests, $p > 0.05$). Data underlying this figure are given in S23 Data.
(TIF)

**S1 Table. Summary of sequencing methods used in this study, and outcomes.**
(DOCX)

**S2 Table. Output of chi-squared tests comparing the proportion of beneficial, neutral, and deleterious mutations across strains in LB.** Values in bold highlight significant differences. Benjamini-Hochberg corrections for multiple comparisons were performed across all tests.
(DOCX)

**S3 Table. Output of chi-squared tests comparing the proportion of beneficial, neutral, and deleterious mutations across strains in glucose.** Values in bold highlight significant differences. Benjamini–Hochberg corrections for multiple comparisons were performed across all tests.
(DOCX)

**S4 Table. Beneficial supply calculations for all strains.** Supply of beneficial mutations ($S_b$) is calculated for each strain in both environments (LB and Glucose) using the empirically estimated $f_b$ values (Fig 4A), whole-genome mutation rates ($\mu$,

Table 1), and genome size (4,641,652 bp) as: $S_b = f_b \times \mu \times$ genome size. Beneficial supply assuming a WT DFE, $S_{b(\text{WT DFE})}$, is calculated as $f_{b(\text{WT})} \times \mu \times$ genome size. $S_b$ and $S_{b(\text{WT DFE})}$ relative to WT are reported in the two rightmost columns. Confidence intervals were calculated as $1.96 \times$ (standard deviation of $S_b$).
(DOCX)

**S5 Table. Deleterious load calculations for all strains.** Deleterious load ($L_d$) is calculated for each strain in both environments (LB and Glucose) using the empirically estimated $f_d$ values (Fig 4A), whole-genome mutation rates ($\mu$) (Table 1), and genome size (4,641,652 bp) as: $L_d = f_d \times \mu \times$ genome size. Deleterious load assuming a WT DFE, $L_{d(\text{WT DFE})}$, is calculated as $f_{d(\text{WT})} \times \mu \times$ genome size. $L_d$ and $L_{d(\text{WT DFE})}$ relative to WT are reported in the two rightmost columns. Confidence intervals were calculated as $1.96 \times$ (standard deviation of $L_d$).
(DOCX)

**S6 Table. Fitness effects of mutations associated with other axes of mutation bias.** The table shows differences between fitness effects of different types of mutations between pairs of strains. Values in bold highlight significant differences.
(DOCX)

**S1 Data. Table with background mutations present in the ancestors of all mutation accumulation (MA) lines.**
(XLSX)

**S2 Data. Tables showing the number and type of mutations present in each MA line.**
(XLSX)

**S3 Data. Table showing raw fitness data for all single-mutation carrying MA lines described in this study, measured in LB and Glucose.**
(XLSX)

**S4 Data.** Data underlying Fig 2.
(XLSX)

**S5 Data.** Data underlying Fig 3.
(XLSX)

**S6 Data.** Data underlying Fig 4.
(XLSX)

**S7 Data.** Data underlying Fig 5.
(XLSX)

**S8 Data.** Data underlying Fig 6.
(XLSX)

**S9 Data. Data underlying S1 Fig.**
(XLSX)

**S10 Data. Data underlying S2 Fig.**
(XLSX)

**S11 Data. Data underlying S3 Fig.**
(XLSX)

**S12 Data.** Data underlying S4 Fig.
(XLSX)

**S13 Data.** Data underlying S5 Fig.
(XLSX)

**S14 Data.** Data underlying S6 Fig.
(XLSX)

**S15 Data.** Data underlying S7 Fig.
(XLSX)

**S16 Data.** Data underlying S8 Fig.
(XLSX)

**S17 Data.** Data underlying S9 Fig.
(XLSX)

**S18 Data.** Data underlying S10 Fig.
(XLSX)

**S19 Data.** Data underlying S11 Fig.
(XLSX)

**S20 Data.** Data underlying S12 Fig.
(XLSX)

**S21 Data.** Data underlying S13 Fig.
(XLSX)

**S22 Data.** Data underlying S14 Fig.
(XLSX)

**S23 Data.** Data underlying S15 Fig.
(XLSX)

**S1 Script.** Custom R script used to align whole-genome sequencing reads to reference, and call mutations from these alignments.
(R)

**S2 Script.** Custom R script used to annotate mutation calls.
(R)

**S3 Script.** Custom Python script used to annotate mutation calls.
(PY)

**S4 Script.** Custom Python script used to make annotated mutation lists.
(PY)

**S5 Script.** Custom R script used to filter annotated mutation calls based on read support.
(R)

**S6 Script. Custom R script used to remove ancestral mutations from mutation lists.**
(R)

**S7 Script. Custom R script used to compile mutation counts and types from mutation lists of individual MA lineages.**
(R)

## Acknowledgments

We thank Pratibha Sanjenbam and Lindi Wahl for critical comments on the manuscript; Lindi Wahl for discussion; Kushan Lahiri for laboratory assistance; and the NGS facility for sequencing.

## Author contributions

**Conceptualization:** Mrudula Sane, Deepa Agashe.

**Data curation:** Mrudula Sane, Shazia Parveen.

**Formal analysis:** Mrudula Sane, Shazia Parveen.

**Funding acquisition:** Deepa Agashe.

**Investigation:** Mrudula Sane, Shazia Parveen, Deepa Agashe.

**Methodology:** Mrudula Sane, Deepa Agashe.

**Project administration:** Deepa Agashe.

**Resources:** Deepa Agashe.

**Supervision:** Deepa Agashe.

**Validation:** Mrudula Sane, Shazia Parveen.

**Visualization:** Mrudula Sane, Shazia Parveen, Deepa Agashe.

**Writing – original draft:** Deepa Agashe.

**Writing – review & editing:** Mrudula Sane, Shazia Parveen, Deepa Agashe.

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
