## [Editor Report · Decision Letter 0]

Dear Deepa,

Thank you for submitting your revised manuscript entitled "Mutation bias alters the distribution of fitness effects of mutations" for consideration as a Research Article by PLOS Biology.

Your revisions have now been evaluated by the PLOS Biology editorial staff and the Academic Editor, and I'm writing to let you know that we would like to send your revised version out for re-review.

Once your full submission is complete, your paper will undergo a series of checks in preparation for peer review. After your manuscript has passed the checks it will be sent out for re-review. To provide the metadata for your submission, please Login to Editorial Manager (https://www.editorialmanager.com/pbiology) within two working days, i.e. by Apr 29 2025 11:59PM.

Kind regards,

Roli

Roland Roberts, PhD

Senior Editor

PLOS Biology

rroberts@plos.org

---

## [Decision Letter · Decision Letter 1]

Dear Deepa,

Thank you for your patience while we considered your revised manuscript "Mutation bias alters the distribution of fitness effects of mutations" for publication as a Research Article at PLOS Biology. This revised version of your manuscript has been evaluated by the PLOS Biology editors, the Academic Editor and two of the original reviewers.

Based on the reviews, we are likely to accept this manuscript for publication, provided you satisfactorily address the remaining points raised by the reviewers. Please also make sure to address the following data and other policy-related requests.

IMPORTANT - please attend to the following:

a) Unfortunately reviewer #3 was unable to re-review; however, we have new comments from both of the others. Reviewer #1 signs off enthusiastically with no further requests. Reviewer #2 concedes that s/he was wrong on one of their previous concerns, but still wants further explanation on the second. S/he also wants you to address some significant presentational issues, and insists (with arguments why) that you include statistical results more prominently in the paper. Please address these remaining requests from reviewer #2.

b) Please address my Data Policy requests below; specifically, we need you to supply the numerical values underlying Figs 2, 3, 4, 5AB, 6ABCD, S1AB, S2, S3, S4, S5, S6, S7, S8, S9, S10, S11ABCD, S12 ABCD, S13ABC, S14, S15ABCDEF, either as a supplementary data file or as a permanent DOI’d deposition. I note that you already supplied the raw data in the existing supplementary data files, but we do also need the values directly underlying each Fig panel.

c) Please cite the location of the data clearly in all relevant main and supplementary Figure legends, e.g. “The data underlying this Figure can be found in S1 Data” or “The data underlying this Figure can be found in https://zenodo.org/records/XXXXXXXX

d) I note that you mention the reviewers ["and three reviewers (including Thomas Bataillon)"] in the Acknowledgements. While we appreciate the sentiment, this is against PLOS policy, so please could you remove it?

e) Please make any custom code available, either as a supplementary file or as part of your data deposition.

We expect to receive your revised manuscript within two weeks.

*Published Peer Review History*

*Press*

Sincerely,

Roli

Roland Roberts, PhD

Senior Editor

rroberts@plos.org

PLOS Biology

DATA POLICY:

Regardless of the method selected, please ensure that you provide the individual numerical values that underlie the summary data displayed in the following figure panels as they are essential for readers to assess your analysis and to reproduce it: Figs 2, 3, 4, 5AB, 6ABCD, S1AB, S2, S3, S4, S5, S6, S7, S8, S9, S10, S11ABCD, S12 ABCD, S13ABC, S14, S15ABCDEF. NOTE: the numerical data provided should include all replicates AND the way in which the plotted mean and errors were derived (it should not present only the mean/average values).

CODE POLICY

DATA NOT SHOWN?

REVIEWERS' COMMENTS:

Reviewer #1:

Firstly, I apologise for the late submission of this review. I continue to really like this paper and think it is an excellent contribution to the field - adding necessary experimental data to a largely theoretical area. The data are robust, and the experimental design is elegant and appropriate to the hypothesis. The authors have responded thoroughly to reviewer comments, and the manuscript is much improved as a result. I particularly appreciated the additional information provided on mutation calling and fitness estimation, which increased my confidence in the reliability of the results. The finding that bias reversal can increase the proportion of beneficial mutations is compelling and well supported. I congratulate the authors on a lovely piece of work.

Reviewer #2:

The authors have presented a revised submission, responding in a detailed way to 3 different reviews, each of which raised multiple points. I appreciate all of their effort to improve the presentation of this very interesting and important study.

My initial review included a critique that, upon consideration, was partly wrong. I argued that mutY should be excluded from any statistical test due to being used previously to argue the same kind of conclusion, and that the non-independence of mutants means that there are only 4 (or 3) data points, not 7. The first point remains valid and it is a simple matter to take this into account (by dropping the data from mutY but keeping mutT and the others). In careful statistical work, we do not purport to test a hypothesis that was proposed based on prior data in a way that includes the same prior data, because the hypothesis is not independent of that data.

However, the second point is not valid. I argued that mutT and mutY were not independent, and likewise that mut S, L and H were not independent, due to hitting the same pathway. This is why I said there were only 4 data points and not 7. But this isn't really correct for the dependent variable, the DFE. The mutations that hit the same pathway are non-independent for the ti/tv bias (and the mutation spectrum generally), but the DFEs they generate are presumably largely independent because (presumably) they are mostly not sharing the same mutations. It would be useful for the authors to explain this if it is, in fact, true.

I still find problems with the presentation. The scatter-plots and correlations in Fig S3, in my opinion, are best suited to convey the main results. I wish that the figures were not so confusing in their use of color. Figure 4 has 7 different color keys. Fig 1 has a colored gradient and also a separate 3-color scheme. Fig 1 needs to present the logic of a prediction (and perhaps the logic of the theory that drives it): when the independent variable (mutation bias) is shifted, this induces (by biased depletion due to the effect of arrival bias) a shift in the dependent variable (DFE). Having actual numbers for ti/tv bias is a distraction.

The authors repeatedly present results in the main text without providing a statistical finding, somethings doing this using words like "significant". They literally have a sentence that says our results support our predictions without specifying the precise substance of either the results or the predictions! In some cases they have done tests, and in other cases they have made a qualitative comparison, e.g., p. 22 line 9 they are using a qualitative analysis in reaching conclusions about Fig. 4 "In contrast, no other axis of variation in mutation spectrum matched the pattern of variation in fb across strains (compare Figure 4A with Figures 4C-F). "

I want to stress that this is a quantitative empirical paper. The statistical results should be front and center, not something that one has to dig to find. Saying "our results support out conclusions" without immediately backing that up with numbers is not satisfactory.

However, having said all that, the paper has been improved in response to the reviews and it does not need to be perfect. Ideally there will be other papers to follow from this group or from others. So, I would recommend the paper for publication without further revision.

---

## [Editor Report · Decision Letter 2]

Dear Deepa,

Thank you for the submission of your revised Research Article "Mutation bias alters the distribution of fitness effects of mutations" for publication in PLOS Biology. On behalf of my colleagues and the Academic Editor, Laurence Hurst, I'm pleased to say that we can in principle accept your manuscript for publication, provided you address any remaining formatting and reporting issues. These will be detailed in an email you should receive within 2-3 business days from our colleagues in the journal operations team; no action is required from you until then. Please note that we will not be able to formally accept your manuscript and schedule it for publication until you have completed any requested changes.

Sincerely, 

Roli

Senior Editor

PLOS Biology

rroberts@plos.org